# Efficacy of Plant-Made Human Recombinant ACE2 against COVID-19 in a Golden Syrian Hamster Model

**DOI:** 10.3390/v15040964

**Published:** 2023-04-14

**Authors:** Choon-Mee Kim, Dong-Min Kim, Mi-Seon Bang, Jun-Won Seo, Da-Young Kim, Na-Ra Yun, Sung-Chul Lim, Ju-Hyung Lee, Eun-Ju Sohn, Hyangju Kang, Kyungmin Min, Bo-Hwa Choi, Sangmin Lee

**Affiliations:** 1Premedical Science, Chosun University College of Medicine, Gwangju 61452, Republic of Korea; 2Department of Internal Medicine, Chosun University College of Medicine, Gwangju 61452, Republic of Korea; 3Department of Pathology, Chosun University College of Medicine, Gwangju 61452, Republic of Korea; 4Department of Preventive Medicine, Jeonbuk National University Medical School, Jeonju 54907, Republic of Korea; 5BioApplications Inc., Pohang Techno Park Complex, 394 Jigok-ro Nam-gu, Pohang 37668, Republic of Korea

**Keywords:** COVID-19, severe acute respiratory syndrome coronavirus 2, angiotensin-converting enzyme 2, Golden Syrian hamster model, treatment efficacy

## Abstract

Coronavirus disease 2019 (COVID-19) is a novel infectious respiratory disease caused by SARS-CoV-2. We evaluated the efficacy of a plant-based human recombinant angiotensin-converting enzyme 2 (hrACE2) and hrACE2-foldon (hrACE2-Fd) protein against COVID-19. In addition, we analyzed the antiviral activity of hrACE2 and hrACE2-Fd against SARS-CoV-2 using real-time reverse-transcription PCR and plaque assays. The therapeutic efficacy was detected using the Golden Syrian hamster model infected with SARS-CoV-2. Both hrACE2 and hrACE2-Fd inhibited SARS-CoV-2 by 50% at concentrations below the maximum plasma concentration, with EC_50_ of 5.8 μg/mL and 6.2 μg/mL, respectively. The hrACE2 and hrACE2-Fd injection groups showed a tendency for decreased viral titers in nasal turbinate tissues on day 3 after virus inoculation; however, this decrease was not detectable in lung tissues. Histopathological examination on day 9 after virus inoculation showed continued inflammation in the SARS-CoV-2 infection group, whereas decreased inflammation was observed in both the hrACE2 and hrACE2-Fd injection groups. No significant changes were observed at other time points. In conclusion, the potential therapeutic efficacy of plant-based proteins, hrACE2 and hrACE2-Fd, against COVID-19 was confirmed in a SARS-CoV-2-inoculated Golden Syrian hamster model. Further preclinical studies on primates and humans are necessary to obtain additional evidence and determine the effectiveness of these therapies.

## 1. Introduction

Coronavirus disease 2019 (COVID-19) is a novel respiratory infectious disease caused by severe acute respiratory syndrome coronavirus 2 (SARS-CoV-2) [1]. Entry of SARS-CoV-2 into host cells depends on the binding of the surface spike (S) protein of the virus to angiotensin-converting enzyme 2 (ACE2) in human cells. The spike glycoprotein of SARS-CoV-2 is a homotrimeric class I fusion protein that undergoes substantial structural rearrangement for binding to ACE2 host cell receptors, subsequently enabling viral entry by membrane fusion [2]. The same mechanism is utilized by the human coronavirus NL63 and SARS coronavirus [3]. According to a previous study, the binding affinity between the soluble ACE2 protein and SARS-CoV S protein, or the equilibrium dissociation constant K_d_, was 1.70 nM, similar to the affinity of human monoclonal antibodies for the S1 protein of SARS-CoV [4].

Attachment of foldon (Fd), a 27-amino-acid phage T4 fibritin, at the C-terminal of a protein can promote trimerization of recombinant fusion proteins or enable the formation of a highly stable protein conformation. In related studies, the receptor binding affinity of influenza virus hemagglutinins for neutralizing the virus was increased using the Fd-trimerization domain [5]. The receptor-binding domain (Fd) trimeric protein created by the fusion of the receptor-binding domain of the Middle East respiratory syndrome coronavirus (MERS-CoV) to Fd protects mice against MERS-CoV infection [6].

Compared to other RNA viruses, coronaviruses mutate less frequently; however, there are recent reports of spreading multiple variants of SARS-CoV-2 (9.8 × 10^−4^ substitutions/site/year) worldwide, leading to a pandemic situation [7]. In particular, mutations in the S protein of SARS-CoV-2 increase the viral infectivity without altering the viral antigenicity, which would have allowed it to evade immune responses brought on by exposure to the original virus or by a vaccine [8]. However, numerous studies have already identified various natural mutations that can lead to escape from monoclonal antibodies [9].

An investigation of ACE2 treatment in mice showed that ACE2 can prevent severe acute lung failure [10]. Furthermore, another study reported that human recombinant angiotensin-converting enzyme 2 (hrACE2) was effective in blocking SARS-CoV-2 infection in both cell culture medium and human organoids [11]. Despite having much scientific evidence of the beneficial effects of hrACE2, to date, no studies have shown a possible therapeutic effect of hrACE2 against COVID-19 in an animal model. Therefore, administering recombinant ACE2 may be an effective therapeutic approach for treating COVID-19 as it not only stops viral spread, but also modifies the renin–angiotensin system to prevent organ damage.

Plants are considered safe and low-cost platforms for the production of various pharmaceutical proteins. In addition, recombinant protein production in plants is compatible with green technologies, has few concerns about pathogens and bacterial endotoxin, is cost-effective, and is widely accepted by society [12,13]. Above all, maximum scale-up possibility, optimized growth procedures, low growth costs, and the possibility of complex protein production suggest that the plant expression system can provide multiple advantages for the production of therapeutic recombinant proteins compared to bacterial or mammalian cell expression platforms [14]. In this study, we evaluated the therapeutic potential of plant-made hrACE2 protein and hrACE2-Fd fusion protein in a Golden Syrian hamster model and SARS-CoV-2-infected Vero E6 cells. 

## 2. Material and Methods

### 2.1. Cell Culture and Virus

The African green monkey kidney Vero E6 cell line (VERO 76) was purchased from the Korean Cell Line Bank (KCLB no. 21587) and maintained in Dulbecco’s modified Eagle’s medium (DMEM, Gibco, Grand Island, NY, USA) supplemented with 10% fetal bovine serum (FBS, Gibco) at 37 °C in a humidified atmosphere of 5% CO_2_. SARS-CoV-2 isolated BetaCoV/South Korea/KUMC01/2020 was propagated in Vero E6 cells. All infection experiments were performed in the biosafety level-3 laboratory of the Health and Environment Research Institute of Gwangju City.

### 2.2. Plant Growth Conditions and Protein Expression 

Plants (*Nicotiana benthamiana*) were grown under a 16:8 h light:dark cycle in a growth room maintained at 25 ± 2 °C and 50 ± 5% relative humidity. The *ace2* and *ace2-Fd* genes were cloned into plant expression vectors. These plant expression vectors were transformed into *A. tumefaciens* strain LBA4404 (TAKARA) using electroporation. Competent cells (100 µL) were mixed with 400 ng of vectors in a 1 mm gap electroporation cuvette (Bio-Rad, Seoul, Republic of Korea). The cells were electroporated using a MicroPulser (Bio-Rad) and stabilized by incubating for 2 h at 28 °C in 1 mL LB broth. Cells were subsequently grown for 2 days at 28 °C on LB agar plates containing 50 µg/mL kanamycin and 25 µg/mL rifampicin as selective antibiotics (Biosesang, Seongnam, Republic of Korea). Transformed agrobacteria were grown for 16 h in 5 mL yeast extract peptone (YEP) liquid medium consisting of 10 g/L yeast extract, 10 g/L peptone, 5 g/L NaCl, 50 mg/L kanamycin, and 25 mg/L rifampicin. Then, 1 mL of cultured cells was inoculated into 1 L of fresh YEP medium and further cultured for 18 h at 28 °C. Cells were then collected by centrifugation at 7341× *g* for 5 min at 4 °C. The harvested cells were resuspended to the desired final OD_600_ in a resuspension solution consisting of 10 mM 2-(N-morpholino) ethanesulfonic acid (Duksan, Ansan, Korea), 10 mM MgCl_2_ (Sigma-Aldrich, Buchs, Switzerland), and 100 mM acetosyringone (Sigma-Aldrich), of pH 5.6. Agroinfiltration was performed by injecting a suspension into the back of the *N. benthamiana* leaves using a vacuum chamber [15].

### 2.3. Protein Purification and Western Blot Analysis

Briefly, 1 kg (fresh weight) of leaves from transgenic plants were harvested and homogenized in a blender (32,000 rpm) in the presence of 2 L of protein extraction buffer (25 mM Tris-Cl, pH 8.0, 300 mM NaCl, 10 mM imidazole, 100 mM sodium sulfite, 0.5% Triton X-100, and 0.5 mM phenylmethylsulfonyl fluoride). To remove debris, the extracts were centrifuged for 40 min at 12,000× *g*, and the supernatants were filtered through Miracloth (EMD Millipore Corp., Billerica MA, USA). The extracts were incubated for 1 h with 100 mL of Ni-IDA agarose resin (Clontech, Kyoto, Japan). The resin was washed with a washing buffer (25 mM Tris-Cl, pH 8.0, 300 mM NaCl, 10 mM imidazole, and 0.5% TWEEN 20). Proteins were eluted by increasing the concentration of imidazole stepwise up to 300 mM. To remove imidazole, the purified protein was exchanged with the final buffer (25 mM Tris-Cl pH 8.0, 300 mM NaCl). The proteins hrACE2 and hrACE2-Fd were stored at 4 °C until further use. Purified proteins were examined using sodium dodecyl sulfate-polyacrylamide gel electrophoresis and Western blot analysis (Appendix A). For Western blot analysis, the proteins were transferred to PVDF (polyvinylidene difluoride) membranes (Merck Millipore Ltd., Tullagreen Carrigtwohill; Ireland) and blocked using by blocking with 5% non-fat dried milk in TBST buffer (20 mM Tris-HCl, pH 7.5, 500 mM NaCl, 0.05% Tween 20) for 30 min at room temperature. The membranes were incubated overnight at 4 °C with a mouse anti-His antibody (Novusbio, Centennial, CO, USA) at a dilution of 1:1000 in TBST buffer containing 1% non-fat dried milk. The membranes were washed three times using TBST buffer, and incubated at room temperature for 1–2 h with 1:5000 diluted HRP-conjugated anti-mouse IgG (H + L) (Bethyl, Montgomery, AL, USA). Finally, the membranes were washed three times using the TBST buffer, and immunoblotted bands were visualized using a SUPEX solution kit (Neutronex, Goryeong, Republic of Korea) and chemiluminescence reagents (Vilber, Collégien, France). Finally, the gels and membranes were stained with Coomassie Brilliant Blue R-250 (Biosolution, Suwon, Republic of Korea).

### 2.4. Antiviral Activity by Real-Time RT-PCR

The antiviral efficacy of hrACE2 against SARS-CoV-2 was quantified by real-time RT-PCR in Vero E6 cells. Briefly, Vero E6 cells were cultured in DMEM supplemented with 10% FBS and 1 × penicillin–streptomycin solution (Gibco, ThermoFisher Scientific Inc., New York, NY, USA) at 37 °C, 5% CO_2_ for 1 d in a 24-well cell culture plates at a density of 2.5 × 10^5^ cells/well. Monolayers of Vero E6 cells in 24-well plates were washed with sterile phosphate-buffered saline (PBS) and then infected with 1 × 10^6^ viral RNA copies/well for 1 h at 37 °C, 5% CO_2_. SARS-CoV-2 was diluted in DMEM, and virus titration was conducted using real-time RT-PCR. The virus was removed and washed three times with PBS. Cells were then treated with two-fold serial dilutions of maximum plasma concentration (Cmax) doses of hrACE2 [16] in DMEM with 10% FBS for 48 h. Cell supernatants were collected for further quantification analysis. Viral RNA was extracted from 200 µL of the supernatant of infected cells using a viral DNA/RNA extraction kit (ZiXpress, Cat No. ZP02201) in an automated nucleic acid purification system (ZiXpress-32), according to the manufacturer’s instructions. The Bioneer Exicycler 96 Real-time PCR kit (Bioneer Inc., Daejeon, Republic of Korea) was used to detect SARS-CoV-2. The nucleocapsid protein (NP) gene was amplified by real-time RT-PCR from the RNA template using the following primers: NP-783F (5′-ACGTACTGCCACTAAAGC-3′), NP-959R (5′-ATGCGCGACATTCCGAAG-3′), and NP-Anti-886P probe (5′-[FAM]TTCCTTGTCTGATTAGTTCCTGGTCC[BHQ1]-3′). A standard curve was generated by determining the copy numbers from serial dilutions (10^1^–10^8^ copies) of the cloned NP gene plasmid. NP DNA (1260 bp) of SARS-CoV-2 was synthesized and cloned into the pBluescript II SK (+) plasmid. The real-time RT-PCR efficiency for the NP gene was 97%, and the R^2^ value was 0.9971. The cycle threshold value was set to ≤39 as the cut-off for positivity. Synthesis of cDNA was as follows: 50 °C for 30 min and 95 °C for 10 min; PCR amplification was performed as follows: 40 cycles at 95 °C for 15 s and 57 °C for 60 s.

### 2.5. Antiviral Activity by Plaque Assay

The antiviral efficacy of hrACE2-Fd against SARS-CoV-2 was quantified using plaque assay in Vero E6 cells. Briefly, Vero E6 cells were cultured in DMEM supplemented with 10% FBS and 1 × penicillin–streptomycin solution at 37 °C, 5% CO_2_ for 1 d in a 24- well cell culture plates at a density of 2.5 × 10^5^ cells/well. Monolayers of Vero E6 cells in 24-well plates were washed with PBS, and then infected with 50–100 plaque-forming units (PFUs)/well SARS-CoV-2 for 1 h at 37 °C, 5% CO_2_. SARS-CoV-2 was diluted in DMEM, and virus titration was conducted using a plaque assay. The virus was removed and washed three times with PBS. Cells were then treated with two-fold serial dilutions of Cmax doses of hrACE2-Fd in DMEM with 5% FBS and 1% methyl cellulose. At the end of the 5-day incubation period, the overlay medium was fixed by adding 1 mL of acetone:methanol (1:1) solution. After washing three times with PBS, crystal violet (1%) solution was added to each well and allowed to stain for 20–30 min. The cell monolayer in each well was rinsed with approximately 1 mL water, and the plates were allowed to air dry. The viral plaques were counted.

### 2.6. Animals and Experimental Hamster Models

Two-month-old male Golden Syrian hamsters (*Mesocricetus auratus*) sourced from Janvier Labs, France, were housed at a specific virus-free certified animal facility. During the acclimatization/stabilization period for the experimental animals, they were randomized by weight into four groups.

The hamsters were subdivided into four groups for different treatments, as presented in Table 1. All the groups were inoculated with 0.1 mL of SARS-CoV-2 (strain NMC-nCoV02) at 10^5^ 50% tissue culture infectious dose (TCID_50_) per milliliter through the nasal cavity. On days 1–5 after the viral inoculation (1–5 days post infection; dpi), 500 µL of either hrACE2 or hrACE2-Fd solution (25 mM Tris-HCl, 150 mM NaCl) was administered via intraperitoneal injection (2.5 mg/(kg body weight) per day) for 5 days in a row (Table 1).

The body weight of each hamster was measured before the infection and daily after viral inoculation, and the average body weight was calculated. Three animals in the PBS control group were dissected on the study end date (at 9 dpi) to measure the viral titer and confirm the presence of lung tissue pathology. Three hamsters each, from the groups consisting of viral infection, ACE2 injection, and ACE2-Fd injection, were dissected at 3, 6, and 9 dpi to measure the viral titer and assess the lung pathology.

### 2.7. Virus Titration in Nasal Turbinate and Lung Tissue Samples

Vero E6 cells were cultured as a monolayer in 96-well cell culture plates to evaluate the viral titer in nasal turbinate and lung tissue samples. The cells were covered with 10-fold serial dilutions (10^0^–10^7^) of nasal turbinate and lung tissue samples in DMEM (Gibco). At the end of a 4–5-day incubation period at 37 °C, 5% CO_2_, the cells were examined for cytopathic effects. Viral titers were calculated using the Reed–Muench method and expressed as log_10_ TCID_50_ per gram [17].

### 2.8. Histopathology

Hamsters were infected with the SARS-CoV-2 strain NMC-nCoV02 at 10^5^ TCID_50_/mL and euthanized at 3, 6, and 9 dpi. For hematoxylin and eosin (H&E) staining, the lungs were excised from each hamster, fixed in 10% neutral formalin, embedded in paraffin, and stained with H&E to analyze pathological changes in the lung tissues. 

### 2.9. Statistical Analysis 

Data on 50% viral plaque reduction relative to virus-only infection (EC_50_) were analyzed with 50% inhibition concentration using GraphPad Prism 8.0.1 (GraphPad Software, San Diego, CA, USA).

Daily changes in body weight were converted to percentages, with body weight on day 0 set to 100%. The Kruskal–Wallis test was performed to compare body weights and viral titers in nasal turbinate and lung tissues between groups. Post hoc analysis involved the Mann–Whitney *U* test, performed when between-group differences were observed. The Bonferroni method was applied to determine the significance of post hoc test results according to the 0.017 (0.05/3) scheme [18]. Statistical analysis was performed using the SPSS (Statistical Package for the Social Sciences) v24.0 software (IBM Corp., Armonk, NY, USA).

## 3. Results

### 3.1. Antiviral Activity of hrACE2 and hrACE2-Fd

The antiviral activity of hrACE2 and hrACE2-Fd was determined by a two-fold serial dilution from the Cmax using real-time RT-PCR or plaque assay. The concentration of hrACE2 and hrACE2-Fd that inhibited SARS-CoV-2 by 50% determined by real-time RT-PCR was an EC_50_ of 5.8 μg/mL and 1.1 μg/mL, respectively (Figure 1A). Additionally, the antiviral activity of hrACE2 and hrACE2-Fd was determined by a plaque assay and showed an EC_50_ of 13.8 μg/mL and 6.2 μg/mL, respectively (Figure 1B). Both hrACE2 and hrACE2-Fd inhibited SARS-CoV-2 by 50% at concentrations below Cmax.

### 3.2. Body Weight Analysis

The experimental design to confirm the therapeutic efficacy of hrACE2 and hrACE2-Fd using the Golden Syrian hamster model infected with SARS-CoV-2 is shown in Figure 2A. Changes in the body weights of hamsters from before infection with SARS-CoV-2 to 9 dpi were measured (Figure 2B). The average body weight in the PBS control group on the last day of the experiment was 5% higher than that at baseline. The viral infection group showed a decrease in body weight starting from 2 dpi. At 4 dpi, the viral infection group showed a 10% decrease in body weight, followed by a continuous increase. The hrACE2 injection group showed a 7% decrease in body weight at 3 dpi, followed by an increase, with a full recovery of body weight at 8 dpi. The hrACE2-Fd injection group exhibited a 7% decrease in body weight at 3 dpi, followed by an increase, with a full recovery of body weight at 8 dpi. There were significant differences between groups at 2, 3, 4, and 5 dpi (*p* value < 0.05 in Kruskal–Wallis test). The results of the post hoc test showed that there were significant differences between the PBS control group and hrACE2 and hrACE2-Fd injection groups at 2 dpi, PBS and viral infection and hrAC2-Fd groups at 3 dpi, and PBS and viral infection groups at 4 and 5 dpi (*p* < 0.017 in Mann–Whitney U test). However, the hrACE2 and hrACE2-Fd groups showed a pattern of weight recovery compared to the virus infection group, although there was no statistical significance for daily body weight changes in hamsters between the viral infection, hrACE2 injection, and hrACE-Fd injection groups. All animals in all groups survived until the end of the experiment (Figure 2B). 

### 3.3. Viral Titers in Lung and Nasal Turbinate Tissues 

After infection with SARS-CoV-2, the viral titer in the hamster nasal turbinate and lung tissues was measured using Vero cells (Figure 3 and Figure 4). Nasal turbinate tissue in the viral infection group showed a viral titer of 4.8–5.8 log_10_TCID_50_/g at 3 dpi and 2.8–3.1 log_10_TCID_50_/g at 6 dpi. No virus was detectable at 9 dpi (limit of detection: 1.8 log_10_TCID_50_/g; Figure 3).

The hrACE2 injection group showed a viral titer of 2.8–3.1 and 1.8–2.1 log_10_TCID_50_/g in the nasal turbinate tissue obtained at 3 and 6 dpi, respectively. The hrACE2-Fd injection group exhibited a viral titer of 4.1–4.8 and 1.8–2.1 log_10_TCID_50_/g in the nasal turbinate tissue obtained at 3 and 6 dpi, respectively. No virus was detectable at 9 dpi (limit of detection: 1.8 log_10_TCID_50_/g; Figure 3). 

Viral titers at 3 dpi were significantly different among the three groups (*p* = 0.034). In the post hoc analysis, the difference between the viral infection and hrACE2 injection groups yielded a *p*-value of 0.043, and that between the viral infection and hrACE2-Fd injection groups was 0.046. Both *p*-values, although greater than 0.017, were lower than 0.05 and, therefore, can be interpreted as a trend (borderline significance). A critical *p* value < 0.017 was considered statistically significant in the post hoc tests. The viral titer at 6 dpi was not significantly different among the three groups. Similar borderline significance was observed in the differences between the viral infection and hrACE2 injection groups (*p* = 0.043) and between the viral infection and hrACE2-Fd injection groups (*p* = 0.046) in the post hoc analysis (Figure 3). 

At 3 dpi, lung tissues in the viral infection, hrACE2 injection, and hrACE2-Fd injection groups contained the virus at concentrations of 5.6–6.1 log_10_TCID_50_/g, 5.1–5.6 log_10_TCID_50_/g, and 5.1–5.8 log_10_TCID_50_/g, respectively. In all three groups, no virus was detectable at 6 and 9 dpi (limit of detection: 1.8 log_10_TCID_50_/g). Therefore, there was no difference in the viral titer in the lung tissues among these groups at 3 and 6 dpi (*p* > 0.05) (Figure 4).

### 3.4. Histopathological Analysis of Lung Tissue in Hamsters Treated with the Experimental Drugs

To assess the treatment efficacy in hamsters who received one of the experimental drugs, histopathological examination of lung tissue was performed at 3, 6, and 9 dpi for each group. The H&E staining of lung tissue collected at 3 dpi revealed an increased number of inflammatory cells in the viral infection group compared with PBS control group (Figure 5A,B). An increased number of inflammatory cells in the lung tissue was also observed in the hrACE2 and hrACE2-Fd injection groups. At 3 dpi, the hrACE2 injection group mainly exhibited mild pneumonic infiltration in the subpleural area, whereas the hrACE2-Fd injection group exhibited pneumonic infiltration at multiple foci (Figure 5E,H). Nevertheless, there were no significant between-group differences in terms of an increase in the number of inflammatory cells compared with the viral infection group (Figure 5).

The H&E staining of lung tissues indicated an overall increase in inflammation in the lung tissue of the virus infection group at 6 dpi compared with that at 3 dpi (Figure 5B,C). Furthermore, the hrACE2 and hrACE2-Fd injection groups showed multifocal pneumonic infiltration at 6 dpi (Figure 5F,I), confirming increased inflammation in comparison with that at 3 dpi (Figure 5E,H). At 6 dpi, both hrACE2 and hrACE2-Fd injection groups generally showed greater inflammation than the viral infection group, although the difference in the presentation of inflammation between the groups was minor (Figure 5F,I).

Compared to that at 6 dpi, at 9 dpi, H&E staining of hamster lung tissue in the viral infection group revealed diffused hemorrhagic pneumonic consolidation (Figure 5C,D, arrows) with pneumonic infiltration (Figure 5C,D, circles) as well as a diffuse increase in the inflammatory cell number in lung tissue, and an increase in alveolar wall thickness, indicating exacerbation of inflammation (Figure 5C,D). Compared to that at 6 dpi, at 9 dpi, the hrACE2 and hrACE2-Fd injection groups showed patchy pneumonic infiltration (circles) without pneumonic consolidation, implying decreased inflammation and inflammatory cell infiltration at 9 dpi (Figure 5G,J). Based on these results, we were able to confirm that pulmonary inflammation due to SARS-CoV-2 led to a continued inflammatory state in the viral infection group, whereas the hrACE2 and hrACE2-Fd injection groups manifested a decrease in inflammation over time (Figure 5). In addition, the histopathological scores of lung tissue lesions among hamster groups are shown in Appendix A.

## 4. Discussion

The renin–angiotensin system is known to play an important role in severe acute lung injury caused by the SARS coronavirus [19]. The binding affinity of SARS-CoV-2 for its receptor ACE2 is 10- to 20-fold stronger than that of SARS-CoV, leading to increased infectivity of SARS-CoV-2 [20]. As not only a functional receptor of coronaviruses but also an important negative regulator of the renin–angiotensin system, ACE2 converts the vasoconstrictor angiotensin II to angiotensin 1-7 and angiotensin I to angiotensin 1-9 [21]. Both ACE2 and angiotensin 1-7 are reported to alleviate endothelial dysfunction and possess cardiovascular protective effects, such as hypotensive, antithrombotic, or antifibrotic activities, as well as lung-protective properties [21,22]. Angiotensin Ⅱ infusion in patients increases the risk of thrombotic events associated with the induction of the proinflammatory cytokine interleukin (IL)-6 [23,24]. In fact, hrACE2 administration in patients with COVID-19 maintains plasma and tracheal ACE2 activity, thereby causing a marked decrease in angiotensin II concentration and an increase in angiotensin 1-7 and 1-9 levels [25]. Furthermore, the downregulation of inflammatory cytokines, such as IL-6, IL-8, and tumor necrosis factor-alpha (TNF-α), which play an important role in lung injury and the cytokine storm, and inflammatory markers, such as C-reactive protein and ferritin, are observed after hrACE2 administration. 

Clinical studies have been conducted on the effectiveness and safety of hrACE2 as a treatment for acute respiratory distress syndrome. It has been demonstrated that hrACE2 treatment does not induce serious adverse events and results in a significant decrease in the concentrations of angiotensin II and IL-6. Furthermore, hrACE2 has been reported to decrease SARS-CoV-2 load by 1000- to 5000-fold in in vitro cell culture experiments and engineered organoid experiments [11]. Such findings have led to a placebo-controlled, double-blind phase 2b trial of hrACE2 in patients with severe COVID-19 [26]; however, no studies on the efficacy of ACE2 have been conducted to date on hamsters. 

In the present study, we confirmed the therapeutic potential of plant-made proteins hrACE2 and hrACE2-Fd against SARS-CoV-2 infection in a hamster model. Hamsters are employed to establish small-animal models of infection with respiratory viruses, and these models feature the expression of receptors in the upper and lower respiratory tract similar to that in humans. Furthermore, the virus receptor and ACE2 amino acid sequences share similarities between humans and hamsters, thereby allowing for analyses of sensitivity to SARS-CoV-1 and SARS-CoV-2 [27]. Therefore, based on this rationale, Golden Syrian hamsters were selected as the animal model for SARS-CoV-2 infection in the present study. In a previous study on REGN-COV2, involving a cocktail of two potent neutralizing antibodies, an in vivo experiment on rhesus macaques showed decreased viral load as well as mild pathological sequelae in the upper and lower airways [28]. Prophylactic administration in a hamster model also revealed a change in the viral RNA copy number in the lungs, but therapeutic administration failed to yield a significant difference compared to the placebo group. Histopathological observations also did not indicate a significant therapeutic effect in the hamster model. 

In the present study, measurement of the viral titer in hamster nasal turbinate tissues showed a decreased viral titer at 3 and 6 dpi in the hrACE2 injection group and at 3 dpi in the hrACE2-Fd injection group, compared to that in the viral infection group (Figure 3 and Figure 4). The H&E staining of hamster lung tissue also revealed a relative improvement in lung lesions in the hrACE2 and hrACE2-Fd injection groups at 9 dpi, although lung lesions and inflammation in either group at 3 and 6 dpi were not significantly different from those in the viral infection group. Therefore, in a hamster model, we confirmed that hrACE2 could reduce the SARS-CoV-2 titer in nasal turbinate tissue as well as alleviate pathological sequelae at 9 dpi, indicating a possible therapeutic action against SARS-CoV-2. Our limitation in this study is that we have not included a quantitative evaluation of the histopathology features observed in the lungs.

In this study, the hrACE2 and hrACE2-Fd proteins only provided very mild protection. Therefore, it is necessary to develop more effective engineered ACE2 proteins. Additionally, further studies are required to determine the blood levels of angiotensin 2 and inflammatory cytokines. Monteil et al. demonstrated that recombinant soluble ACE2 enhanced the effect of remdesivir on Vero E6 and kidney organoids in combinatorial regimens for COVID-19 treatment [29]. Therefore, more enhanced hrACE2 and antiviral agent combination regimens for COVID-19 treatment experiments are required in primates or human subjects regarding recombinant ACE2 and its effect on SARS-CoV-2. 

We demonstrated that hrACE2 and hrACE2-Fd inhibited SARS-CoV-2 at concentrations below Cmax in the Vero E6 cell line in vitro. Several SARS-CoV-2 variants have been identified since the beginning of the COVID-19 pandemic, and dominantly circulating variants include B.1.1.7 (alpha), B.1.351 (beta), P.1 (gamma), B.1.617.2 (delta), C.37 (lambda), B.1.1.529 (omicron), and BA.4 or BA.5 (omicron variants) (https://www.who.int/emergencies/diseases/novel-coronavirus-2019 (accessed on 06 April 2023)). Because hrACE2 shows a tendency to reduce the wild-type SARS-CoV-2 viral load in nasal turbinate tissue, hrACE2 may perform better against the ongoing pandemic omicron variants which have intrinsically higher replication competence in the upper respiratory tract (nasal and bronchi tissues tropism). Therefore, further studies are needed to elucidate the therapeutic efficacy of hrACE2 and hrACE2-Fd against these dominantly circulating SARS-CoV-2 variants.

Our study results suggest that hrACE2 acts as a soluble receptor for SARS-CoV-2; it exerts an antiviral effect, as observed with neutralizing antibodies, and thus would have reduced the viral load in hamster nasal turbinate tissues. Another likely explanation for the beneficial effect of hrACE2 against SAR-CoV-2 is that the conversion of angiotensin II to angiotensin 1-7 by hrACE2 may lead to anti-inflammatory effects owing to the blockade of angiotensin II-induced oxidative stress, which may be associated with a relative improvement in lung lesions in the hrACE2 and hrACE2-Fd injection groups at 9 dpi [30]. This study has certain limitations. Angiotensin Ⅱ levels were not measured in hamster blood; thus, further studies are required to demonstrate hrACE2-induced changes in angiotensin II levels and the renin-angiotensin system. In addition, studies are needed to determine the pharmacokinetics and pharmacodynamics of these proteins to elucidate the therapeutic activity of hrACE2 and hrACE2-Fd in the lungs, nasal cavity, and blood. Moreover, further studies are required to determine blood levels of inflammatory cytokines, such as IL-6, IL-8, IL-1β, and TNF-α, that play an important role in lung injury.

In conclusion, using a Golden Syrian hamster model and Vero E6 cells in vitro, this study demonstrated the potential therapeutic efficacy of hrACE2 protein and hrACE2-Fd fusion protein against COVID-19. Further preclinical studies on primates and human subjects are required to obtain additional evidence and confirm the effectiveness of these therapies.

## Figures and Tables

**Figure 1 viruses-15-00964-f001:**
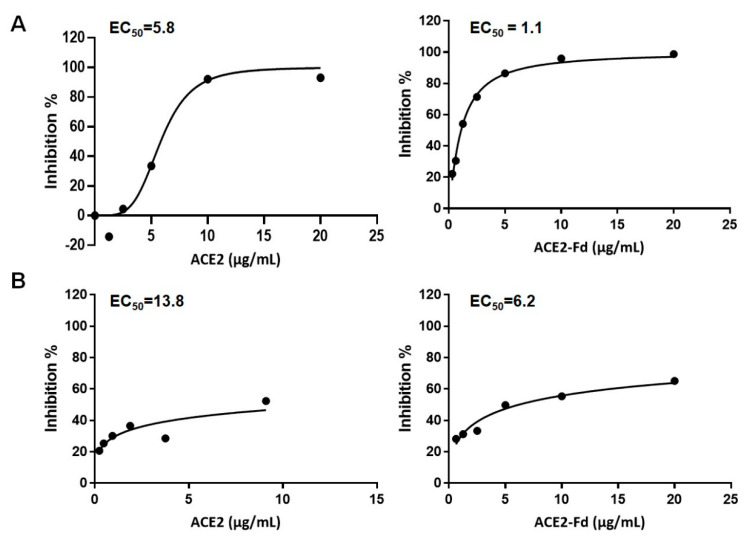
Antiviral activity of hrACE2 and hrACE2-Fd against SARS-CoV-2 in Vero E6 cells. (**A**) Antiviral efficacy of hrACE2 and hrACE2-Fd against SARS-CoV-2 was quantified using real-time RT-PCR. (**B**) Antiviral efficacy of hrACE2 and hrACE2-Fd against SARS-CoV-2 was quantified using plaque assay. hrACE2, human recombinant angiotensin-converting enzyme 2; Fd, foldon; SARS-CoV-2, severe acute respiratory syndrome coronavirus 2.

**Figure 2 viruses-15-00964-f002:**
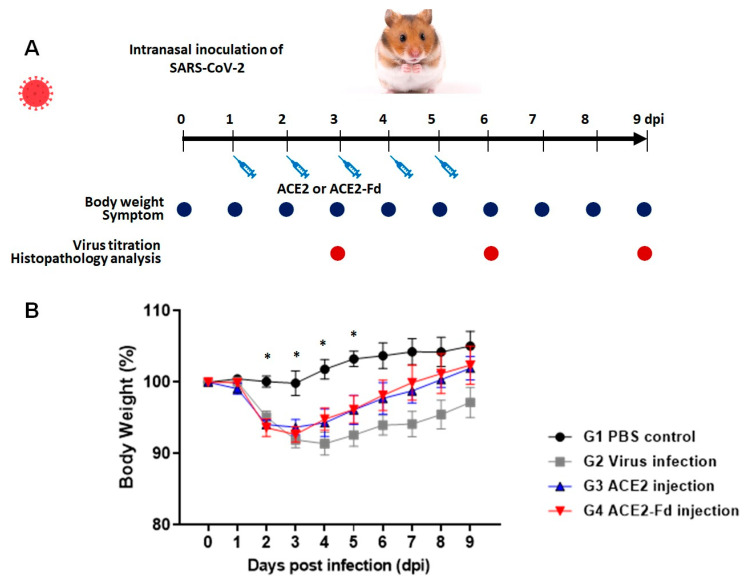
Therapeutic effect of hrACE2 or hrACE2-Fd on SARS-CoV-2 in Golden Syrian hamsters. (**A**) Schematic overview of this study. (**B**) Changes in the body weight of hamsters. Hamsters were intranasally inoculated with 100 µL of 10^5^ TCID_50_/mL SARS-CoV-2, and 2.5 mg/kg of hrACE2 or 2.5 mg/kg of hrACE2-Fd recombinant protein was intraperitoneally injected for 5 days (1–5 dpi). Changes in the body weight of hamsters were monitored from before virus infection (day 0) to 9 dpi. Means (±standard error of the means (SEMs)) are shown for each group. ACE2, angiotensin-converting enzyme 2; Fd, foldon; PBS, phosphate-buffered saline. PBS control (G1), virus infection (G2), hrACE2 injection (G3), and hrACE2-Fd injection (G4). Significant differences (*p* < 0.05) determined using the Kruskal–Wallis test are shown as asterisks (*), and multiple comparisons using the Mann–Whitney U test were G1 vs. G3, G4 at 2 dpi (*p* = 0.009, 0.006), G1 vs. G2, G4 at 3 dpi (*p* = 0.007, 0.012), G1 vs. G2 at 4 dpi (*p* = 0.004), and G1 vs. G2 at 5 dpi (*p* = 0.004).

**Figure 3 viruses-15-00964-f003:**
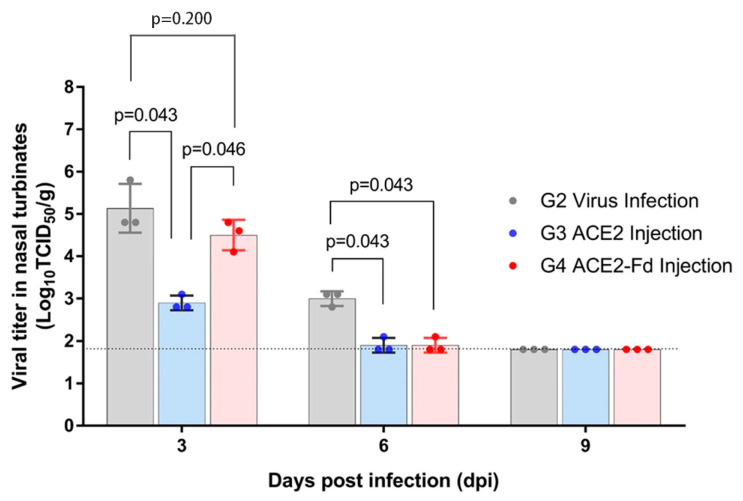
Viral titer in nasal turbinate tissue. Hamsters were intranasally inoculated with 100 µL of 10^5^ TCID_50_/mL SARS-CoV-2, and 2.5 mg/kg of hrACE2 or 2.5 mg/kg of hrACE2-Fd recombinant protein was intraperitoneally injected for 5 days (1–5 dpi). Viral titer in nasal turbinate tissue was measured by TCID_50_ using Vero E6 cells at 3, 6, and 9 dpi. ACE2, angiotensin-converting enzyme 2; Fd, foldon; TCID_50_, 50% tissue culture infectious dose.

**Figure 4 viruses-15-00964-f004:**
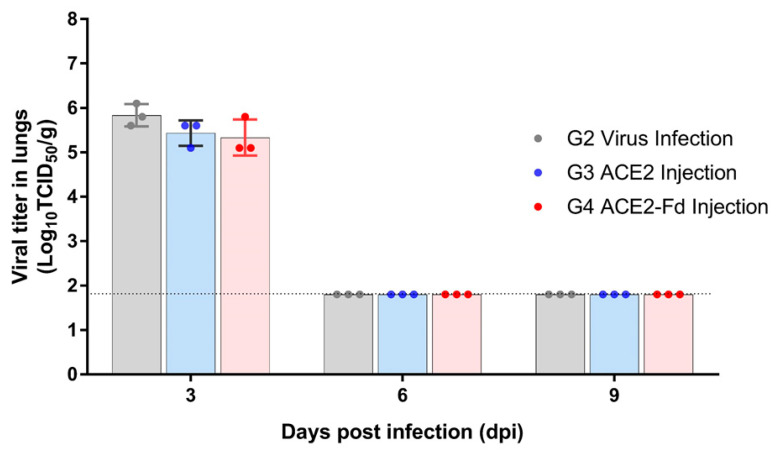
Viral titer in the lung tissue. Hamsters were intranasally inoculated with 100 µL of 10^5^ TCID_50_/mL SARS-CoV-2, and 2.5 mg/kg of hrACE2 or 2.5 mg/kg of hrACE2-Fd recombinant protein was intraperitoneally injected for 5 days (1–5 dpi). Viral titer in lung tissue was measured by TCID_50_ using Vero E6 cells at 3, 6, and 9 dpi. ACE2, angiotensin-converting enzyme 2; Fd, foldon; TCID_50_, 50% tissue culture infectious dose.

**Figure 5 viruses-15-00964-f005:**
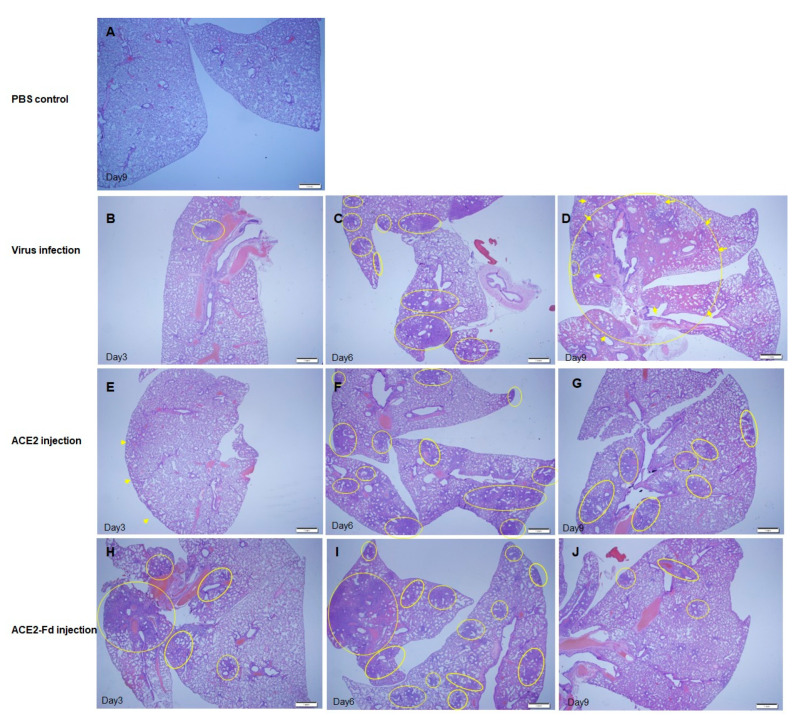
Comparison of lung tissue lesions among hamster groups at different time points. (**A**) PBS control group, Day 9: hematoxylin and eosin (H&E) staining. (**B**) Virus infection group, Day 3: histopathological finding of an area of pneumonic infiltration (circle) with H&E staining. (**C**) Virus infection group, Day 6: histopathological findings of multiple foci of pneumonic infiltration (circles) with H&E staining. (**D**) Virus infection group, Day 9: histopathological findings of diffuse hemorrhagic pneumonic consolidation (arrows) with pneumonic infiltration (circles) with H&E staining. (**E**) hrACE2 injection group, Day 3: histopathological findings of mild pneumonic infiltration predominantly along the subpleural area (arrows) with H&E staining. hrACE2: human recombinant angiotensin-converting enzyme 2. (**F**) hrACE2 injection group, Day 6: histopathological findings of multiple foci of pneumonic infiltration (circles) with H&E staining. (**G**) hrACE2 injection group, Day 9: histopathological findings of patchy pneumonic infiltration (circles) without pneumonic consolidation, with H&E staining. (**H**) hrACE2-Fd injection group, Day 3: Histopathological findings of multiple foci of pneumonic infiltration (circles) with H&E staining. Fd: foldon. (**I**) hrACE2-Fd injection group, Day 6: histopathological findings of multiple foci of pneumonic infiltration (circles) with H&E staining. (**J**) hrACE2-Fd injection group, Day 9: histopathological findings of patchy pneumonic infiltration (circles) without pneumonic consolidation, with H&E staining.

**Table 1 viruses-15-00964-t001:** Experiment schedule in Golden Syrian hamster model infected with SARS-CoV-2 to determine hrACE2 and hrACE2-Fd therapeutic efficacy.

Groups	Hamster No.	Challenge Dosage and Strains	Drug Administration and Injection Volume	Monitoring
1. Control group	3	Intranasal inoculation (100 µL of PBS)	-	Monitoring of weight change for 9 days after challenge, dissection at 9 dpi
2. Virus infection group	9	Intranasal inoculation (100 µL of 10^5^ TCID_50_/mL), SARS-CoV-2 (strain NMC-nCoV02)	-	Monitoring of weight change for 9 days after challenge, virus titration and histopathology analysis after dissection at 3, 6, and 9 dpi
3. ACE2 injection group	9	Intranasal inoculation (100 µL of 10^5^ TCID_50_/mL), SARS-CoV-2 (strain NMC-nCoV02)	Intraperitoneal injection of 500 µL of hrACE2 solution (2.5 mg/(kg body weight) per day) on days 1–5 after the viral inoculation (1–5 dpi)	Monitoring of weight change for 9 days after challenge, virus titration and histopathology analysis after dissection at 3, 6, and 9 dpi
4. ACE2-Fd injection group	9	Intranasal inoculation (100 µL of 10^5^ TCID_50_/mL), SARS-CoV-2 (strain NMC-nCoV02)	Intraperitoneal injection of 500 µL of ACE2-Fd solution (2.5 mg/(kg body weight) per day) on days 1–5 after the viral inoculation	Monitoring of weight change for 9 days after challenge, virus titration and histopathology analysis after dissection at 3, 6, and 9 dpi

## Data Availability

All relevant data are within the manuscript.

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
