# Peer review of "Efficacy of Plant-Made Human Recombinant ACE2 against COVID-19 in a Golden Syrian Hamster Model"

_viruses, 2023, doi:10.3390/v15040964_

Round 1

Reviewer 1 Report

Comments on “Efficacy of Plant-made Human Recombinant ACE2 against

COVID-19 in a Syrian Hamster Model”

  The paper by Kim et al have evaluated the anti-SARS-CoV-2 efficacy of plant-made human recombinant ACE2 both in Vero E6 cells and hamster model. Pandemic COVID-19 has resulted in millions of deaths and enormous economic loss around world. To fight against the pandemic, numerous antiviral strategies have been evaluated. However, the various mutations found in VOCs have already identified can led to escape from the existing neutralizing antibodies. Thus, soluble protein hACE2, the functional receptor of SARS-CoV-2, has shown it’s therapeutic potential for all existing and the upcoming variant strains. Authors evaluated antiviral activity of a plant-based hrACE2 (human recombinant ACE2) and hrACE2-fd (hrACE2-folden) protein. The EC50 of these two proteins show microgram level in Vero E6 cells. However, both hrACE2 and hrACE2-fd show little protective efficacy in hamster after 104 PFU of i.n. infection with following 5 doses of hACE2 i.p. injection. Although the results indicate the decrease of viral titers in nasal turbinate on day 3 and the improvement of lung injury on day 9, it’s tough to say whether the soluble hACE2 is useful as an antiviral countermeasure against COVID-19. There are some comments below for author’s consideration:

1>   To quantify the antiviral (especially neutralizing activity) activity of a candidate drug in Vero E6 cells, researchers usually perform the virus neutralizing procedures before infection. Why did authors treat cells after virus infection.

2>   Authors mentioned “there were significant differences between PBS control group vs hrACE2-Fd injection group in day 2, PBS vs viral infection group in day 8 and 9” in chapter 3.2. Is there any significance between virus infection group and ACE2 injection groups? Which may provide evidence for its therapeutic potentials. Besides, please mark the significance on Fig.2b.

3>   The EC50 of hrACE2-fd in Vero cells is lower than hrACE2, however, the in vivo antiviral activity (in nasal turbinate) of hrACE2 show more obvious than that of hrACE2-fd. What’s the possible reason? Besides, check the description in chapter 3.3, paragraph 3, “and that between the viral infection and hrACE2-Fd injection groups was 0.046”.

4>     While hACE2 injection showed a tendency for decreased virus titer in nasal turbinate, what’s the possible reason for its “no difference in the viral titer in the lung tissues among these groups at 3 and 6 dpi”? Authors may further determine the pharmacokinetics of these two proteins in lung, nasal and blood, which may explain the “ineffectiveness”.

5>   The histopathological pictures show that hrACE2-fd injection result in more severe lung injury on day 3 but less lung lesion on day 9 when compared with the other two groups, it’s hard to say “Based on these results, we were able to confirm that pulmonary inflammation due to SARS-CoV-2 led to a continued inflammatory state in the viral infection group, whereas the hrACE2 and hrACE2-Fd injection groups manifested a decrease in inflammation over time” (page 13), it’s better to introduce a pathological scoring system and marking every single lung. Besides, check the format of the figure legend in Fig.5.

6>   In order to better explain the therapeutic activity of hrACE2, authors should further determine the blood concentration of some common inflammatory cytokines such as IL-1b, IL-6…

7>   As authors showed in this manuscript, hrACE2 show a tendency for decreased wild type SARS-CoV-2 strain titer in nasal turbinate, hrACE2 may play a better performance against ongoing pandemic Omicron variant, a “nose tropism” strain.

Reviewer 2 Report

SARS-CoV-2 was identified as an infectious respiratory disease in December 2019. Vaccines and drugs have been developed and used to prevent and treat the disease.  Choon-Mee Kim and colleagues developed planted-made human recombinant ACE2 protein, hrACE2 and hrACE2-Fd and tested the antiviral activity using qRT-PCR and plaque assay. Golden Syrian hamster challenge showed very mild protection compared to the control group. Overall, the study is well-designed, and the experimental approaches and the interpretation of the data are appropriate. But there are several points to be strengthened to make such conclusions. Some major comments/questions are found below:

1.    In the introduction part, the authors did not claim why to choose plant-sourced proteins. What made the authors think it’s better than mammalian cells?

2.    The authors used betaCoV, the very early strain to test the protection role. The authors need to explore the protection in the recent strain.

3.    Please show the date of the expression and purification date of protein hrACE2 and hrACE2-Fd.

4.    For the statistic in Figure 2B, please indicate the statistical results for G2 vs G3, and G2 vs G4.

5.    The authors also need to include the H&E date for bronchus.

6.    Histopathological scores of inflammations were needed to provide for Figure 5.

7.    The authors did not discuss well why the proteins only provided very mild protections.

8.    What do the authors think in figure 3, the ACE2-Fd group showed a higher titer than the ACE2 group on Day 3.

Minor comments:

9.     Please use full name to indicate: Golden Syrian Hamster

10. The labeling for Figure 5 in the Results part is very confusing. (not in right order)

Round 2

Reviewer 1 Report

no more comments

Reviewer 2 Report

Thanks to the authors for explaining my concern. The article is ready to go.